# Synthesis of *meta*-carbonyl phenols and anilines

Bao-Yin Zhao[1], Qiong Jia[1] & Yong-Qiang Wang [1]✉

Phenols and anilines are of extreme importance for medicinal chemistry and material science. The development of efficient approaches to prepare both compounds has thus long been a vital research topic. The utility of phenols and anilines directly reflects the identity and pattern of substituents on the benzenoid ring. Electrophilic substitutions remain among the most powerful synthetic methods to substituted phenols and anilines, yet in principle achieving *ortho*- and *para*-substituted products. Therefore, the selective preparation of *meta*-substituted phenols and anilines is the most significant challenge. We herein report an efficient copper-catalyzed dehydrogenation strategy to exclusively synthesize *meta*-carbonyl phenols and anilines from carbonyl substituted cyclohexanes. Mechanistic studies indicate that this transformation undergoes a copper-catalyzed dehydrogenation/allylic hydroxylation or amination/oxidative dehydrogenation/aromatization cascade process.

Phenols and anilines are core structures of pharmaceuticals, agrochemicals, polymers, dyestuffs and natural products[1–4]. The development of efficient approaches to preparing both compounds has thus long been a vital research topic. Current strategies to synthesize phenols mainly include: (1) direct oxidation of the C(sp²)−H bond[5–10] (Fig. 1a(i)); (2) transition metals-assisted conversion of the C(sp²)−X bond[11–15] (Fig. 1a(ii)); (3) oxidation of the C(sp²)−M bond[16,17] (Fig. 1a(iii)). In 2011, Stahl et al. reported an elegant palladium-catalyzed aerobic dehydrogenation of substituted cyclohexanones to phenols[18] (Fig. 1a(iv)); and recently Tu et al. developed a dehydrogenation-coupling protocol to synthesize *ortho*-substituted phenols via cyclohexanones from cyclohexanols[19]. Compared to phenols, the synthesis of anilines from a corresponding C(sp²)−H bond involves multistep sequences and/or harsh conditions. For instance, it was first converted into nitro, nitroso, azo, azide, imine, or amide/imide/sulfonamide, followed by one or more additional steps to arrive at an amine[20] (Fig. 1a(v)). Thanks to the development of metal-promoted reactions for over a century, three great name reactions, namely Ullmann−Goldberg reaction[21,22] (Fig. 1a(vi)), Buchwald−Hartwig reaction[23–25] (Fig. 1a(vii)), and Chan−Lam reaction[26–28] (Fig. 1a(viii)) have been sequentially explored for the formation of C(sp²)-N bonds. These reactions have become fundamental tools for the preparation of

anilines in numerous areas of basic and applied research[29,30]. However, all of these procedures are restricted by the need for a pre-functionalized arene, such as aryl halides, aryl pseudohalides, aryl organoborons, etc. Here, we present an alternative approach for the synthesis of phenols and anilines from carbonyl-substituted cyclohexanes (Fig. 1a(ix)).

The utility of phenols and anilines directly reflects the identity and pattern of substituents on the benzenoid ring. Therefore, the introduction of chemical functional groups on the specific site of their benzenoid ring represents a key point. Electrophilic aromatic substitutions (S_EAr) are classical chemical reactions that remain among the most powerful synthetic methods to substituted phenols and anilines, in principle achieving *ortho*- and *para*-substituted products due to the strong electron-donating nature and directing effects of the hydroxyl group and amino group. While the innate site selectivity limits the production of *meta*-substituted phenols and anilines, it yet has been inspiring extensive efforts of synthetic chemists to break through the limitation. In the past decades, the transition metal−catalyzed C−H activation strategy has provided a significant access to this challenge; and these methods mainly relied on: ruthenium-catalyzed σ-bond activation, remote directing template introduction, noncovalent interaction (e.g., hydrogen-bonding and ion-pair interaction), norbornene and a

[1]Key Laboratory of Synthetic and Natural Functional Molecule Chemistry of Ministry of Education, College of Chemistry & Materials Science, School of Foreign Languages, Northwest University, Xi'an 710069, China. ✉e-mail: wangyq@nwu.edu.cn

**Fig. 1 | Synthesis of *meta*-carbonyl phenols and anilines. a** Synthesis of phenols and anilines. **b** Recent work. **c** Proposed reaction process.

carboxylic acid group traceless assistance, steric hindrance effects, etc.[31] Despite their high efficiency for *meta*-C(sp²)–H functionalization of phenols and anilines, these approaches required numerous functional group additions or manipulations in order to furnish the desired product. Furthermore, all of these methods inevitably generated a certain ration of *ortho*- and *para*-isomers; *ortho*-, *meta*-, and *para*-isomers always possess similar polarity and physical properties, thus their separation is always difficult. And it is known that the purification of the product is a key issue in industrial manufacturing. Herein, we disclose a specific synthesis of *meta*-substituted phenols and anilines by a copper-catalyzed dehydrogenation-allylic oxidation or amination-dehydrogenative aromatization sequence without *ortho*- and *para*-isomers detected. In the protocol, the Cu-catalyst plays crucial roles not only in promoting the dehydrogenation/allylic hydroxylation/oxidative dehydrogenation/aromatization cascade processes, but also in controlling the chemo- and regioselectivities.

Inspired by recent work on the synthesis of 1,4-enediones from saturated ketones[32] (Fig. 1b) and related literature[33,34], we hypothesized that a similar process might proceed when cyclohexyl took the place of the alkyl chains moiety of ketones, and if that was followed by further dehydrogenation and aromatization, *meta*-functionalized phenols would be obtained (Fig. 1c). However, the cascade transformation is challenging in view of the previous report where the cyclic ketone substrate generated a fully dehydrogenative aromatized product under dehydrogenation conditions[35].

## Results

### Reaction development

The initial investigation began with cyclohexyl(phenyl)methanone **1a** as the model substrate in the presence of Pd(OAc)₂ (10 mol%), Cu(OAc)₂ (10 mol%) and trifluoroacetic acid (TFA, 10.0 equiv) in DMSO (0.25 M) at 90 °C under O₂ for 60 h (Table 1). As expected, the desired *meta*-carbonyl product **2a** was obtained, albeit in a low yield of 6% (entry 1). We also predictably detected the formation of small amounts of completely dehydrogenative aromatized byproduct **5**, as well as

*para*-functionalized product **6** in the reaction, which motivated us to turn our attention to improving the yield and chemo- and regioselectivities of the reaction. Then with Pd(OAc)₂ as the catalyst, the other copper salts were investigated. The desired product **2a** was obtained in 16% yield, with almost negligible product **6**, when CuI was employed as the co-catalyst (entry 2). Replacement of CuI with other copper salts led to inferior results (entry 3). In light of these results, we assumed that the CuI catalyst might have a vital impact on the yield and selectivity of the reaction. Indeed, the targeted **2a** was delivered in 41% yield in the presence of CuI without Pd catalyst, whereas omission of CuI only led to a small quantity of **2a**, indicating that CuI was key catalyst to convert **1a** into the corresponding *meta*-carbonyl product **2a** (entries 4 and 5). Further, we investigated other metal (e.g., Ni, Fe, Co and Ag) salts as co-catalysts or oxidants to improve the yield of **2a** (entries 6–10). The experiments revealed that most Ag salts could contribute to high yields, while other metal salts proved ineffective (entries 9 and 10), and AgOAc emerged as the most productive co-catalyst for the reaction, providing the targeted product **2a** in 53% yield (entry 9). Note that the ratio of AgOAc to CuI was crucial for the reaction. It was found that a larger amount of AgOAc inhibited the reactivity, while AgOAc (9 mol%) provided **2a** in 64% yield (entry 11). Furthermore, when H₂O (50 μL, 11.1 equiv) as an additive was introduced into the reaction system, **3** and **4** were completely consumed, improving the yield to 76% (entry 12). This result implies that water additive accelerates the transformation of **3** and **4** into **2a** and also suggests that **3** and **4** were likely reaction intermediates to produce **2a** (Supplementary Information Section 5.4). Moreover, the use of ¹⁸O-labeled water provided the desired product with ¹⁸O incorporation, showing the hydroxyl introduced into the products did come from the water[36] (Supplementary Fig. 5). A series of control experiments revealed that all the parameters in the reaction system were essential for the reaction to proceed; omission of either CuI or TFA resulted in no reaction (entries 11, 13–16). Alternatively, the reaction could also proceed in excellent yield under silver-free conditions, when using *tert*-butyl hydroperoxide (TBHP) instead of AgOAc and O₂ (entry 17).

**Table 1 | Optimization of reaction conditions[a]**

| Entry | Catalyst | Co-catalyst/Oxidant | 2a | 3 | 4 | 5 | 6 |
|---|---|---|---|---|---|---|---|
| **Yield (%)[b]** | | | | | | | |
| 1 | Pd(OAc)₂ | Cu(OAc)₂ | 6 | trace | trace | 11 | 7 |
| 2 | Pd(OAc)₂ | CuI | 16 | trace | trace | 7 | trace |
| 3[c] | Pd(OAc)₂ | Other Cu salts | <8 | trace | trace | <17 | <10 |
| 4 | Pd(OAc)₂ | - | trace | trace | trace | 12 | 5 |
| 5 | - | CuI | 41 | trace | trace | <5 | n.d. |
| 6[d] | CuI | Ni salts | <39 | <11 | trace | <5 | n.d. |
| 7[e] | CuI | Fe salts | <32 | <9 | <12 | <5 | n.d. |
| 8[f] | CuI | Co salts | <24 | <9 | trace | <5 | n.d. |
| 9 | CuI | AgOAc | 53 | 17 | 13 | 6 | n.d. |
| 10[g] | CuI | Other Ag salts | <51 | <20 | <17 | <8 | n.d. |
| 11[h] | CuI | AgOAc | 64 | 15 | 8 | <5 | n.d. |
| 12[i] | CuI | AgOAc | 76 (71)[j] | trace | trace | <5 | n.d. |
| 13 | CuI | - | 54 | trace | trace | <5 | n.d. |
| 14 | - | AgOAc | - | - | - | - | - |
| 15[k] | CuI | AgOAc | - | - | - | - | - |
| 16[l] | CuI | AgOAc | 68 | trace | 8 | <5 | n.d. |
| 17[m] | CuI | TBHP | 73 | trace | trace | <5 | n.d. |

*TBHP* tert-butyl hydroperoxide. *n.d.* not detected.

[a]Reaction conditions: **1a** (0.25 mmol), Catalyst (10 mol%), Oxidant (10 mol%), and TFA (10.0 equiv) in DMSO (1 mL) under O₂ at 90 °C for 60 h.

[b]The yields were determined by 1H NMR using dibromomethane as the internal standard.

[c]Other Cu salts see Supplementary Table 1.

[d]Ni salts see Supplementary Table 2.

[e]Fe salts see Supplementary Table 3.

[f]Co salts see Supplementary Table 4.

[g]Other Ag salts see Supplementary Table 5.

[h]9 mol% AgOAc was used.

[i]50 µL H₂O was added.

[j]Isolated yield is shown in parentheses.

[k]Without TFA.

[l]Under air.

[m]TBHP (2.2 equiv) instead of AgOAc and O₂.

## Substrate scope

Under the optimized conditions, we subsequently studied the substrate scope of the reaction (Fig. 2a). A wide range of aryl ketones bearing various substituent groups all performed well, affording the corresponding *meta*-carbonyl phenols in good to excellent yields (**2a–2r**). Substrates containing electron-donating (**2b, 2c, 2h–2j**, and **2n-2q**) and electron-withdrawing (**2d–2g, 2k–2m**, and **2r**) groups on the aromatic ring were all well compatible, delivering the corresponding products in 51 to 88% yield. Halogens such as fluorine (**2d, 2k**, and **2r**), chlorine (**2e** and **2l**), bromine (**2f**) and iodine (**2g**) were tolerated, giving the desired products in good yields. Di- and tri-substituted aromatic substrates also reacted smoothly under the standard reaction conditions to generate the corresponding products (**2s-2v**). In addition, other aromatic or heterocyclic substrates such as naphthyl, furyl, thienyl, and benzothienyl were also applicable, as exemplified with products **2y, 2z, 2aa**, and **2ab**, furnished in up to 81% yield (**2y–2ab**). The structure of **2k** was confirmed by single-crystal X-ray diffraction (Supplementary Information Section 6). The scalability of this reaction was investigated by a 5.5 mmol scale reaction of cyclohexyl(phenyl)methanone **1a**, in which the desired product **2a**

could be obtained in 65% isolated yield (Supplementary Information Section 3.3).

Notably, conjugated enones **7** were also suitable for this transformation, producing bioactive chalcone derivatives in high yields (**8a-8l**), which further extends the diversity of this reaction (Fig. 2b). Functional groups such as methyl (**8b, 8e**, and **8g**), methoxy (**8h**), methylthio (**8i**), alkynyl (**8d**), and chlorine (**8c** and **8f**) were well tolerated, giving the *meta*-carbonyl phenols in 53–83% yield. Additionally, heterocyclic substrates were also well behaved with high efficiency (**8k** and **8l**). Interestingly, in addition to conjugated enones, conjugated dienones were also amenable to the standard conditions, generating products **8m** to **8r** in excellent yields regardless of their electronic properties. The structure of **8a** was confirmed by single-crystal X-ray diffraction (Supplementary Information Section 6).

To further expand the scope of this reaction, we next investigated other aliphatic ketones and aldehydes **9** (Fig. 2c). By slightly tuning the reaction conditions, norfenefrine and etilefrine precursor **10a** could be successfully synthesized. The cyclopropane ring, a highly reactive group, remained intact in the reaction, demonstrating the reaction conditions were mild. It should be mentioned that the product from

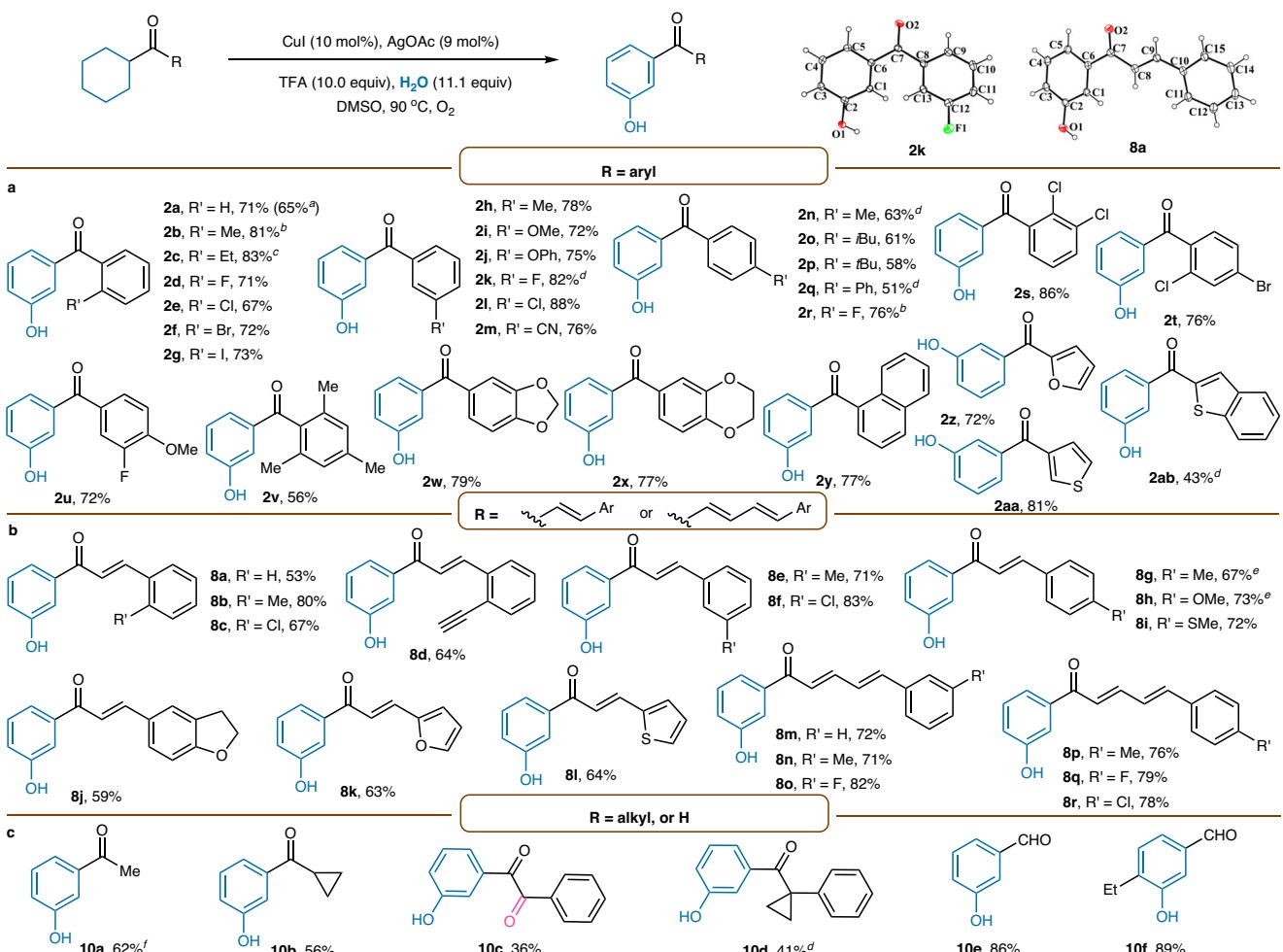

**Fig. 2 | Substrate scope for synthesis of *meta*-carbonyl phenols. a** Substrate scope of aryl ketones. **b** Substrate scope of conjugated enones. **c** Substrate scope of aliphatic ketones and aldehydes. Reaction conditions: substrate (0.25 mmol), CuI (10 mol%), AgOAc (9 mol%), and TFA (10.0 equiv) in DMSO (1 mL), H₂O (50 μL) under O₂ at 90 °C for 60 h. Isolated yields are reported. [a]Reaction conducted with 1a (5.5 mmol). [b]AgOAc (9.5 mol%) was used. [c]48 h. [d]100 °C. [e]No addition of H₂O. [f]Pd(OAc)₂ (15 mol%) was used instead of AgOAc.

ketone with α-methylene C−H bonds provided benzyl-oxygenated *meta*-carbonyl phenol **10c**. The aliphatic ketone with α-quaternary center was amenable to the oxidative conditions, giving the expected product **10d**. Of note, cyclic aldehydes, which are sensitive to oxidizing conditions, were also compatible with this transformation and furnished their corresponding products in high yields (**10e** and **10f**). It is well-known that ketone and aldehyde could be converted into various other groups, such as alkane, alkene, alcohol and acid, thus various *meta*-substituted phenols could be synthesized by this approach (Supplementary Information Section 5.7).

Buoyed by the success in the synthesis of *meta*-carbonyl phenols, we turned our focus on the synthesis of *meta*-carbonyl anilines. As mentioned above, the hydroxyl group of *meta*-carbonyl phenols came from water in this reaction system. It is reasonable that, instead of water, the introduction of amine into the reaction system would produce *meta*-carbonyl anilines. Thus, we investigated the reaction between cyclohexyl(phenyl)methanone **1a** and aniline **11a** under the standard reaction conditions. As anticipated, the expected *meta*-carbonyl aniline **12a** was obtained, along with a small amount of *meta*-carbonyl phenol **2a**, presumably because of a certain amount of water introduced by hygroscopic DMSO. As aforementioned, the reaction could also proceed with an excellent yield when *tert*-butyl hydroperoxide (TBHP) took the place of AgOAc and O₂. After extensive screening of the reaction conditions (see Supplementary Tables 9−12

for optimization details), the desired *meta*-carbonyl aniline **12a** was delivered in 73% yield in the presence of CuI (10 mol%), TBHP (2.2 equiv) and trifluoroacetic acid (TFA, 10.0 equiv) in DMSO (0.25 M) at 90 °C for 60 h.

With optimal conditions in hand, we subsequently evaluated the scope of anilines **11** by using cyclohexyl(phenyl)methanone **1a** as a model partner (Fig. 3a). Anilines bearing electron-donating or electron-withdrawing groups were feasible, producing desired *meta*-carbonyl diphenylamines in 57−85% yield (**12a-12p**). Substituents at the *otho*, *meta*, or *para* positions were amenable to the catalytic system. Various functional groups, including Me, OMe, *i*Pr, *t*Bu, OPh, F, Cl, Br, and I, were also found to be suitable for this transformation. In addition, multi-substituted anilines also proved applicable under the standard conditions, furnishing the *meta*-carbonyl anilines in up to 89% yield (**12q-12w**). Gratifyingly, the 4-methoxyphenyl (PMP) group in **12m** could be easily removed via the oxidation with cerium(IV) ammonium nitrate to deliver primary amine product (Supplementary Information Section 5.7). The structures of **12i** and **12k** were unambiguously confirmed by single-crystal X-ray diffraction (Supplementary Information Section 6).

We further surveyed the scope of aromatic ketones with aniline **11a** (Fig. 3b). Substrates bearing substituents with distinct electron properties on the aromatic ring moiety were feasible, delivering the corresponding products in moderate to good yields. Additionally,

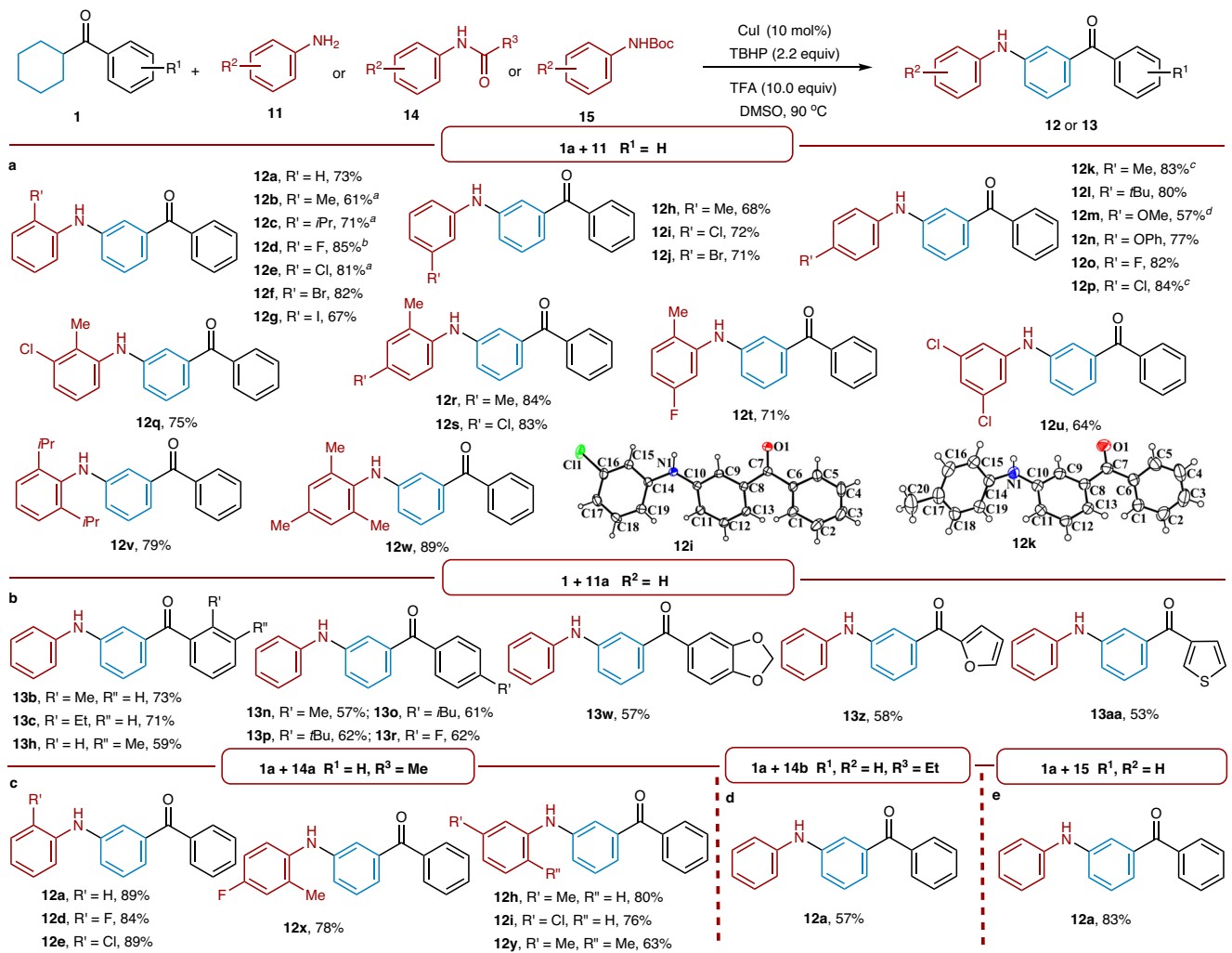

**Fig. 3 | Substrate scope for synthesis of *meta*-carbonyl anilines. a** Substrate scope of anilines. **b** Substrate scope of aromatic ketones. **c** Substrate scope of *N*-acetyl anilines. **d** Substrate of *N*-propionyl aniline. **e** Substrate of *N-tert*-butoxycarbonyl aniline. Reaction conditions: **1** (0.25 mmol), **11** (0.5 mmol) or **14** (0.5 mmol) or **15** (0.5 mmol), CuI (10 mol%), TBHP (2.2 equiv), and TFA (10.0 equiv) in DMSO (1 mL) at 90 °C for 60 h. Isolated yields are reported. ᵃTBHP (1.1 equiv). ᵇTBHP (2.0 equiv). ᶜ36 h. ᵈAgOAc (9 mol%) instead of TBHP. Boc, *tert*-butoxycarbonyl.

substituents at various positions were also viable for this strategy. Furthermore, heterocyclic substrates such as furyl and thienyl all occurred smoothly, also giving the expected products in 53–58% yield (**13z** and **13aa**).

Encouraged by the foregoing success with anilines, we wondered whether the corresponding *meta*-carbonyl anilines could be obtained when using protected anilines instead of anilines. We then tested the reaction between protected anilines **14** and **15** and cyclohexyl(phenyl) methanone **1a** (Fig. 3c–e). Unexpectedly, protected anilines including *N*-acetyl aniline **14a**, *N*-propionyl aniline **14b** and *N-tert*-butoxycarbonyl aniline **15a** all furnished the same deprotected product **12a** in 89%, 57%, and 83% yields, respectively. *N*-acetyl anilines with electronically diverse substituents all proceeded smoothly, giving the *meta*-carbonyl deprotected anilines in good to excellent yields. In view of these results, we presumed that the protected anilines could be more stable than unprotected anilines under the oxidative conditions. Disubstituted substrates were also found to be compatible with the standard conditions, producing the desired products in good yields (**12x** and **12y**).

## Mechanistic investigation
To gain insight into the reaction mechanism, a series of isotope tracking studies, deuterium-labeling experiments and kinetic studies

were conducted. Initially, the ¹⁸O-labeled water was introduced into the reaction system of **1a** to probe the origin of the hydroxyl (Fig. 4a). As a consequence, the ¹⁸O-labeled product **2a** was obtained by high-resolution mass spectrometry (HRMS) analysis, suggesting the hydroxyl in phenol did come from the water (Supplementary Fig. 5). Next, we carried out deuterium labeling experiments of **1a** with $D_2O$ under the standard reaction conditions, providing the deuterium-labeling product with 48% D incorporation at the $\gamma'$ position. In addition, the recovered **1a** was found to contain 50% D at the $\alpha$ position (Fig. 4b). The occurrence of H/D exchange indicates that this catalytic system enables the activation of the $\alpha$-C–H and $\gamma$-C–H bond. Additionally, the kinetic isotope effect (KIE) experiments showed the value of $k_H/k_D = 1.38$ for parallel experiments of **1a** and **1a**-*d*, indicating that the cleavage of the $\alpha$-C–H bond may not be involved in the rate-limiting step (Fig. 4c). Subsequently, intermediate trapping experiments were conducted using substrate **1a** to shed light onto the reaction mechanism (Fig. 4d). In addition to the desired product **2a**, compounds **3**, **4**, and **16** were also observed and isolated. To gain deeper understanding into the reaction process, we conducted a kinetic time course analysis of the reaction using ¹HNMR. Monitoring the reaction process via periodic sampling revealed that $\alpha,\beta$-unsaturated ketone **3** is initially formed, allylation product **4** subsequently formed, and 1,4-enedione **16** ultimately formed. Compounds **3**, **4** and

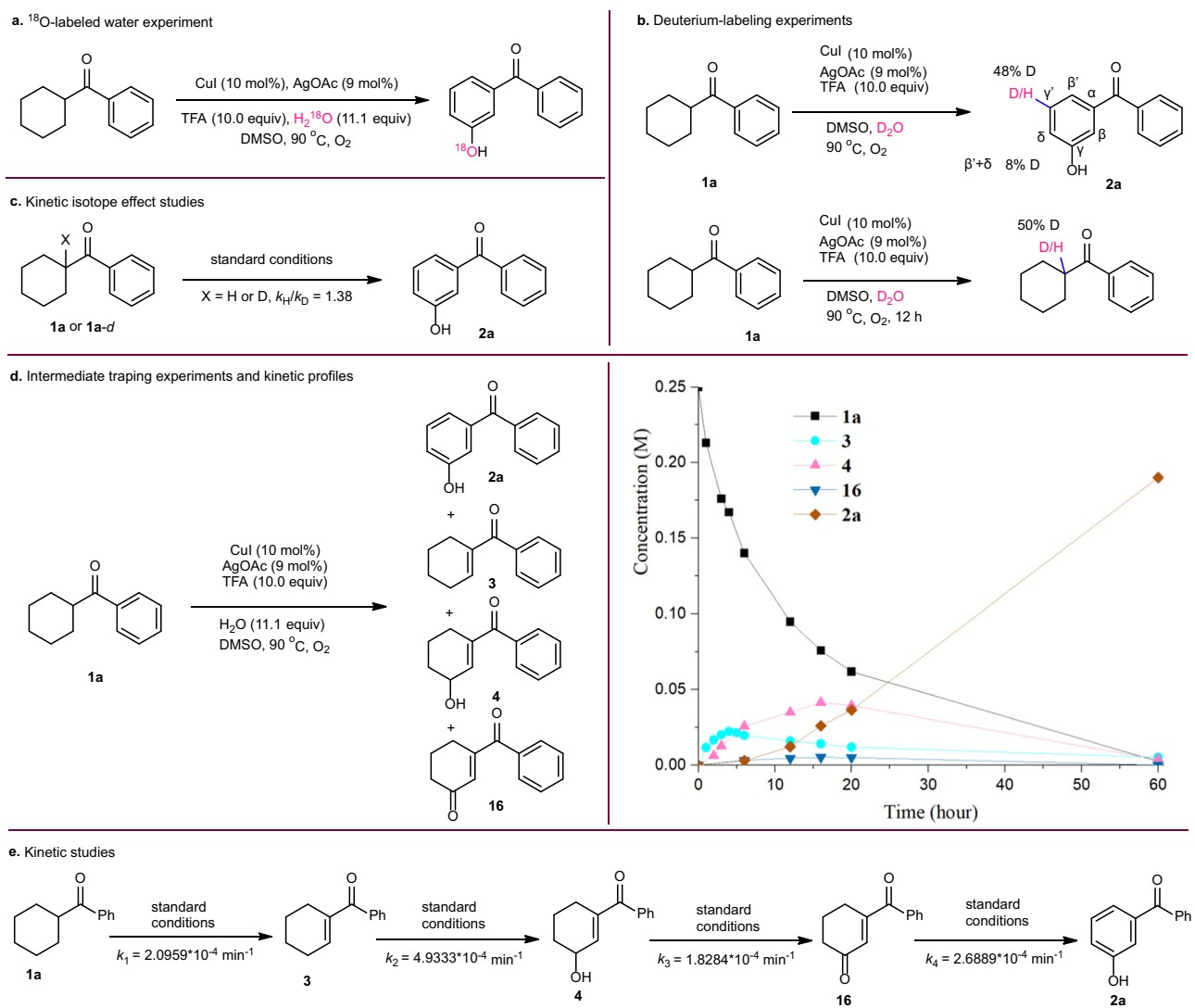

**Fig. 4 | Mechanistic investigation. a** [18]O-labeled water experiment. **b** Deuterium-labeling experiments. **c** Kinetic isotope effect studies. **d** Intermediate traping experiments and kinetic profiles. **e** Kinetic studies.

**16** are subsequently consumed as the product **2a** is formed. These results provided evidence to support that **3**, **4** and **16** are reaction intermediates to produce the product **2a**. We reasoned that synthesis of *meta*-carbonyl phenols proceeds through the following cascade (**1a** → **3** → **4** → **16** → **2a**). To obtain kinetic rate constants of each step, the substrate **1a** and intermediates **3**, **4** and **16** were subjected to the standard conditions, respectively (Fig. 4e). The rate constants for each step were determined as follows: $k_1 = 2.0959 \times 10^{-4}\,min^{-1}$, $k_2 = 4.9333 \times 10^{-4}\,min^{-1}$, $k_3 = 1.8284 \times 10^{-4}\,min^{-1}$, $k_4 = 2.6889 \times 10^{-4}\,min^{-1}$, giving a ratio of $k_1/k_2/k_3/k_4 = 1.15{:}2.70{:}1.00{:}1.47$. These results indicated that the allylic oxidation of **4** to **16** was the rate-limiting step. Based on the experimental results and the related literature[37–39], a plausible reaction mechanism for synthesis of *meta*-carbonyl phenols or anilines was proposed (Supplementary Fig. 23).

## Discussion

In conclusion, we have established an efficient synthetic protocol for *meta*-substituted phenols and anilines. The approach is simple, proceeds under mild conditions, and uses abundant and inexpensive copper catalysts, rendering it practical advantages. Furthermore, this transformation is of complete *meta*-site selectivity, and no *ortho*- and/or *para*-isomers were detected, which facilitates

subsequent purification processes. We anticipate that this strategy will be useful for the synthesis of other *meta*-functionalized aromatic compounds.

## Methods

### General procedure for the synthesis of *meta*-carbonyl phenols

A 15 mL sealed tube containing a magnetic stir bar was charged with ketones (0.25 mmol), CuI (4.8 mg, 10 mol%), AgOAc (3.7 mg, 9 mol%), and DMSO (1 mL) sequentially. The tube was evacuated and backfilled with $O_2$ three times. Subsequently, $H_2O$ (50 μL, 11.1 equiv) and TFA (186 μL, 10.0 equiv) were added. The tube was sealed and the mixture was stirred at 90 °C for 60 h. After cooling to room temperature, the reaction mixture was diluted with EtOAc (50 mL) and washed with $H_2O$ (3 × 1 mL) and brine (3 × 1 mL). The combined organic layer was dried over anhydrous $Na_2SO_4$, and concentrated to give the residue, which was further purified by silica gel column chromatography to afford the desired product.

### General procedure for the synthesis of *meta*-carbonyl anilines

A 15 mL sealed tube containing a magnetic stir bar was charged with ketones (0.25 mmol), anilines (0.5 mmol), CuI (4.8 mg, 10 mol%), DMSO (1 mL) and TFA (186 μL, 10.0 equiv) sequentially. Subsequently,

TBHP (100 µL, 2.2 equiv) was added. The tube was sealed and the mixture was stirred at 90 °C for 60 h. After cooling to room temperature, the reaction mixture was diluted with EtOAc (50 mL) and washed with $H_2O$ (3 × 1 mL) and brine (3 × 1 mL). The combined organic layer was dried over anhydrous $Na_2SO_4$, and concentrated to give the residue, which was further purified by silica gel column chromatography to afford the desired product.

## Data availability

The data supporting the findings of this study are available within the article and its Supplementary Information. Additional data are available from the corresponding author upon request. Crystallographic data for the structures reported in this Article have been deposited at the Cambridge Crystallographic Data Centre, under deposition numbers CCDC 2264711 (2k), 2264714 (8a), 2264715 (12i) and 2264716 (12k). Copies of the data can be obtained free of charge via https://www.ccdc.cam.ac.uk/structures/.

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

## Acknowledgements

We are grateful for financial support from the National Natural Science Foundation of China (No. NSFC-22371228 and 21971206 for Y.-Q.W.). We thank X.-Q. Wang for assistance with acquiring HRMS data. We thank the

X-ray Crystallography Facility for X-ray crystallography. We also thank L. Yang for the X-ray crystallographic analysis of compounds **2k**, **8a**, **12i**, and **12k**.

## Author contributions

Y.-Q.W. conceived the study. B.-Y.Z. discovered and conducted experiments. Y.-Q.W. proposed the mechanism. B.-Y.Z. conducted preliminary mechanistic studies. Y.-Q.W., Q.J. and B.-Y.Z. wrote the manuscript. Y.-Q.W. directed the project.

## Competing interests

The authors declare no competing interests.
