## [Peer Review File · Nature Communications]

Reviewers' Comments:

Reviewer #1:

Remarks to the Author:

The manuscript presents a copper-catalyzed dehydrogenation strategy to exclusively synthesize meta-functionalized phenols and anilines from carbonyl-substituted cyclohexanes. The approach is simple, selective, uses inexpensive copper catalysts, and avoids the subsequent purification processes, which renders its practical advantages. However, the substrate scope is restricted to only carbonyl functionalized cyclohexanes, which decreases the generality and significance of the protocol. More importantly, only several basic control experiments are performed. Consequently, some key insightful explanations are missing. For instance, why are only pure meta-substitution products observed without any ortho- and para-products? What are the key factors behind the excellent selectivity? How does the carbonyl group affect the reaction? As a directing group? More experimental and computational studies should be carried out. This reviewer thinks the manuscript lacks deeper understanding of the mentioned new protocol and does not provide enough inspiring take-home messages. Therefore, the manuscript is not suitable for publishing in Nat. Comm. in its current form.

Reviewer #2:

Remarks to the Author:

The authors have provided satisfactory responses to the earlier comments. I recommend proceeding with the publication of the manuscript.

Reviewer #3:

Remarks to the Author:

This manuscript is an improved version of a manuscript that I have previously reviewed for another journal. The results are certainly impressive and the mechanistic implications for the field of copper catalysis are profound. My concerns have been addressed by the revisions. I have no further comments on the paper, which is well prepared.

Response Letter

Reviewer #1 (Remarks to the Author):

“The manuscript presents a copper-catalyzed dehydrogenation strategy to exclusively synthesize *meta*-functionalized phenols and anilines from carbonyl-substituted cyclohexanes. The approach is simple, selective, uses inexpensive copper catalysts, and avoids the subsequent purification processes, which renders its practical advantages. However, the substrate scope is restricted to only carbonyl functionalized cyclohexanes, which decreases the generality and significance of the protocol. More importantly, only several basic control experiments are performed. Consequently, some key insightful explanations are missing. For instance, why are only pure *meta*-substitution products observed without any *ortho*- and *para*-products? What are the key factors behind the excellent selectivity? How does the carbonyl group affect the reaction? As a directing group? More experimental and computational studies should be carried out. This reviewer thinks the manuscript lacks deeper understanding of the mentioned new protocol and does not provide enough inspiring take-home messages. Therefore, the manuscript is not suitable for publishing in Nat. Comm. in its current form.”

Response: We thank the reviewer very much for the valuable comments.

As suggested, more mechanistic experiments were carried out.

1. Based on the experimental results and the relevant references, below is our explanation why our reaction system favors the formation of *meta*-products.

1) We conducted intermediate species isolating experiments under the standard reaction conditions with **1a** as the substrate. To our delight, intermediate products **3**, **4**, and **16** were isolated (Figure 1r). To gain deeper understanding into the reaction process, we conducted a kinetic time course analysis of the reaction using ¹HNMR. Monitoring the reaction process revealed that α,β -unsaturated ketone **3** is initially formed, allylation product **4** subsequently formed, and 1,4-enedione **16** ultimately

formed. Compounds **3**, **4** and **16** are subsequently consumed as the product **2a** is formed. These results provided evidence to support that **3**, **4** and **16** are reaction intermediates to produce the product **2a**. We reasoned that synthesis of *meta*-carbonyl phenols proceeds through the following cascade (**1a**→**3**→**4**→**16**→**2a**). To obtain kinetic rate constants of each step, the substrate **1a** and intermediates **3**, **4** and **16** were subjected to the standard conditions, respectively. The rate constants for each step were determined as follows: $k_1 = 2.0959 \times 10^{-4} \text{ min}^{-1}$, $k_2 = 4.9333 \times 10^{-4} \text{ min}^{-1}$, $k_3 = 1.8284 \times 10^{-4} \text{ min}^{-1}$, $k_4 = 2.6889 \times 10^{-4} \text{ min}^{-1}$, giving a ratio of $k_1/k_2/k_3/k_4 = 1.15 : 2.70 : 1.00 : 1.47$.

Figure 1r. Intermediate trapping experiments, kinetic profiles and reaction process.

2) The selectivity of *ortho*-, *meta*- or *para*-products is determined by the step of the oxidation of intermediates **3** to **4**. As shown in Figure 2r, because the carbon of carbonyl group possesses the partial positive charge, so α -position and γ -position of carbonyl group possesses the partial negative charge, while β -position and β' -position possess the partial positive charge. Therefore, γ -position is easier oxidized than β -position and β' -position. From ^1H NMR spectrum and ^1H - ^1H COSY NMR spectrum of compound **3** (figure 3r), the proton of γ position has lower chemical shift than the proton of β' position, which also indicates that the γ position is more electron-rich and easier oxidized than β' -position.

Figure 2r. The oxidation of intermediates **3** to **4**.

Figure 3r. ^1H NMR and ^1H - ^1H COSY NMR spectra of **3** (400 MHz, CDCl_3)

3) As shown in Figure 4r, there are main three pathways to generate *ortho*-product. Path i is allylic oxidation of β' -position. However, as mentioned above, γ -position is easier oxidized than β -position and β' -position. In this case, our reaction system favors the formation of *meta*-products. Path ii involves a Michael addition/oxidation/further dehydrogenation cascade process. However, Michael addition is typically under the basic condition, which does not match our acid reaction conditions. In this case, our reaction system disfavors the formation of *ortho*-products. The related example of base-assisted Michael addition reactions are reported (*Curr. Org. Chem.* **26**, 1264-1293 (2022)). Path iii is the direct oxidation of α,β -unsaturated ketone **3** to 1,3-diketone. Generally, the oxidation hardly proceeded due to the electrical property.

Figure 4r. The pathway to generate *ortho*-product.

4) To obtain *para*-product, the homoallylic oxidation of δ -position must be proceeded. As we known, homoallylic oxidation is less reactive than allylic oxidation, thus homoallylic oxidation is hard to achieve under the reaction conditions.

2. To our delight, beside compounds **2a**, **3**, **4** and **16** (Figure 5r), the key intermediate species **F** and **H** were also detected by high-resolution mass spectrometry (HRMS) analysis. For intermediate **F**: HRMS (ESI) m/z : $[M + Na]^+$ Calcd for $C_{15}H_{13}F_3O_3CuNa$ 384.0005; Found 383.9997 (Figure 6r); For intermediates **H**: HRMS (ESI) m/z : $[M + Na]^+$ Calcd for $C_{15}H_{13}F_3O_3Na$ 321.0709; Found 321.0694 (Figure 7r).

Figure 5r. The intermediates detected in reaction system of **1a**.

Figure 6r. HRMS spectrum of the key intermediate species **F**.

Figure 7r. HRMS spectrum of the key intermediate species **H**.

Based on the experimental results and the related literatures (*J. Am. Chem. Soc.* **141**, 14889–14897 (2019); *J. Am. Chem. Soc.* **131**, 5044–5045 (2009); *J. Am. Chem. Soc.* **133**, 15300–15303 (2011).), a plausible reaction mechanism for synthesis of *meta*-carbonyl phenols was proposed as figure 8r (which could also be found in Supplementary Information as Supplementary Figure 23.). First, Cu^I species is oxidized in situ by AgOAc in the presence of TFA under oxygen atmosphere, generating Cu^{II}(O₂CCF₃)₂; meanwhile, the ketone suffers from the enolization. The formation of copper(II) enolate followed by the oxidation or the disproportionation gives copper(III) enolate that undergoes β-hydride elimination to deliver the α,β-unsaturated ketone **3** along with a Cu^{III}-hydride intermediate. The Cu^{III}-hydride species eliminates a TFA, resulting in Cu^IO₂CCF₃ that is reoxidized to Cu^{II}(O₂CCF₃)₂ by AgOAc and O₂ in the presence of TFA. Subsequently, α,β-unsaturated ketone **3** isomerizes into diene **E**. The terminal C=C double bond of diene is activated by Cu^{II}(O₂CCF₃)₂, and then delivers into Cu^{II} species **F**, meanwhile losing a TFA. Cu^{II} species **F** can be detected by HRMS (HRMS (ESI) m/z: [M + Na]⁺ Calcd for C₁₅H₁₃F₃O₃CuNa 384.0005; Found 383.9997). Cu^{II} species **F** undergoes the oxidation

or the disproportionation to give Cu^{III} intermediate **G** which proceeds a reductive elimination to generate intermediate **H** and $\text{Cu}^{\text{I}}\text{O}_2\text{CCF}_3$ that proceeds the same process as above to regenerate the active $\text{Cu}^{\text{II}}(\text{O}_2\text{CCF}_3)_2$. Intermediate **H** can also be detected by HRMS (HRMS (ESI) m/z : $[\text{M} + \text{Na}]^+$ Calcd for $\text{C}_{15}\text{H}_{13}\text{F}_3\text{O}_3\text{Na}$ 321.0709; Found 321.0694). Then intermediate **H** is hydrolyzed to furnish the product **4** (*J. Org. Chem.* **86**, 7603–7608 (2021)). **4** is oxidized producing 1,4-enedione **16** which then undergoes the similar procedure like **1a** to **3**, affording the targeted product **2a**. The related examples of Cu^{II} disproportionation to give the Cu^{III} intermediate are reported (*Angew. Chem. Int. Ed.* **50**, 11062–11087 (2011); *J. Am. Chem. Soc.* **131**, 5044–5045 (2009)).

Figure 8r. Proposed reaction mechanism for synthesis of *meta*-carbonyl phenols.

3. In our reaction system, the carbonyl group might not act as a directing group role. Actually, the carbonyl group enables the activation of the α -position and γ -position by the formation of enole **A** and diene **E** (Figure 8r). In mechanistic investigation section, we carried out deuterium labeling experiments of **1a** with D_2O under the standard reaction conditions, giving the deuterium-labeling product with 48%

D incorporation at the γ' position. Besides, the recovered **1a** was found to contain 50% D at the α position. Please see figures 9r and 10r.

Figure 9r. ^1H NMR spectrum of compound deuterium-labeling **2a** (400 MHz, CDCl_3)

Figure 10r. ^1H NMR spectrum of the recovered **1a** (400 MHz, CDCl_3)

Reviewer #2 (Remarks to the Author):

“The authors have provided satisfactory responses to the earlier comments. I recommend proceeding with the publication of the manuscript.”

Response: We thank the reviewer very much for supporting the publication of this work in Nature Communications!

Reviewer #3 (Remarks to the Author):

“This manuscript is an improved version of a manuscript that I have previously reviewed for another journal. The results are certainly impressive and the mechanistic implications for the field of copper catalysis are profound. My concerns have been addressed by the revisions. I have no further comments on the paper, which is well prepared.”

Response: We thank the reviewer for supporting the publication of this work in Nature Communications!

+++

Reviewer #1:

My concerns have been well addressed in the response. I recommend further proceeding for publication.